# Fault Diagnosis of Power Transformer Based on Time-Shift Multiscale Bubble Entropy and Stochastic Configuration Network

**DOI:** 10.3390/e24081135

**Published:** 2022-08-16

**Authors:** Fei Chen, Wanfu Tian, Liyao Zhang, Jiazheng Li, Chen Ding, Diyi Chen, Weiyu Wang, Fengjiao Wu, Bin Wang

**Affiliations:** 1Department of Power and Electrical Engineering, Northwest A&F University, Xianyang 712100, China; 2Key Laboratory of Agricultural Soil and Water Engineering in Arid and Semiarid Areas of Ministry of Education, Northwest A&F University, Xianyang 712100, China; 3Yellow River Engineering Consulting Co., Ltd., Zhengzhou 450003, China; 4Wuling Power Corporation Ltd., Changsha 410004, China

**Keywords:** power transformer, fault diagnosis, multiscale entropy, stochastic configuration networks, feature extraction

## Abstract

In order to accurately diagnose the fault type of power transformer, this paper proposes a transformer fault diagnosis method based on the combination of time-shift multiscale bubble entropy (TSMBE) and stochastic configuration network (SCN). Firstly, bubble entropy is introduced to overcome the shortcomings of traditional entropy models that rely too heavily on hyperparameters. Secondly, on the basis of bubble entropy, a tool for measuring signal complexity, TSMBE, is proposed. Then, the TSMBE of the transformer vibration signal is extracted as a fault feature. Finally, the fault feature is inputted into the stochastic configuration network model to achieve an accurate identification of different transformer state signals. The proposed method was applied to real power transformer fault cases, and the research results showed that TSMBE-SCN achieved 99.01%, 99.1%, 99.11%, 99.11%, 99.14% and 99.02% of the diagnostic rates under different folding numbers, respectively, compared with conventional diagnostic models MBE-SCN, TSMSE-SCN, MSE-SCN, TSMDE-SCN and MDE-SCN. This comparison shows that TSMBE-SCN has a strong competitive advantage, which verifies that the proposed method has a good diagnostic effect. This study provides a new method for power transformer fault diagnosis, which has good reference value.

## 1. Introduction

The power transformer is an important part of the power grid, and ensuring the safe operation of the transformer is a prerequisite for the stability of the power system [1,2]. Therefore, the effective detection of potential transformer faults based on the transformer monitoring system and fault diagnosis technology, has important engineering value for maintaining the efficient operation of the power grid [3,4,5].

As an emerging transformer detection method, the online detection method overcomes the defect of the traditional offline detection method, which requires a power fault [6]. It has the advantages of convenience, safety and reliability, and meets power supply requirements. It has become the mainstream direction of power transformer fault diagnosis [7,8]. Generally speaking, transformer online detection methods can be summarized into two categories: dissolved gas analysis (DGA) and vibration analysis [9]. DGA [10,11,12,13] is mainly based on the dissolved gas composition of transformer oil, and diagnostic technology to detect faults caused by the change of insulating oil properties, such as partial discharge and overheating. For example, Wu et al. [11] proposed a DGA transformer fault diagnosis method based on the combination of improved seagull optimization algorithm and support vector machine, which can accurately identify transformer faults such as overheating and discharge. However, the diagnostic effect of DGA is deeply affected by the concentration of dissolved gas, and detection equipment is expensive, resulting in DGA having certain limitations. As the external manifestation of power transformers, mechanical vibration signals contain a large amount of information about the operation state of the transformers [14]. Therefore, researchers often use vibration signals as an important indicator to evaluate the health status of the transformer, and transformer fault diagnosis based on vibration signals has been widely used [15,16,17]. However, most of the existing research on transformer fault diagnosis based on vibration signals focuses on mechanical faults such as internal windings [18] and iron core loosening [19]. They ignore the effective detection of over-excitation, under-excitation and inter-turn short circuit, which may cause serious consequences such as transformer heating, and even, burning. Therefore, it is necessary to reasonably detect the abnormal conditions of the transformer under different excitation states, including bad states such as winding inter-turn short circuit.

The transformer fault diagnosis method based on vibration signal mainly includes two steps: feature extraction and pattern recognition [20,21]. Feature extraction is the key to transformer fault diagnosis, and the quality of feature extraction directly affects the final diagnosis results. As a nonlinear dynamic method, entropy measures the complexity of the signal. It is widely used in the field of rotating machinery fault feature extraction such as rolling bearing [22,23], wind turbine [24,25] and hydropower units [26,27]. At the same time, with the continuous development of entropy theory, isentropic models similar to multiscale entropy [28], refined composite multiscale entropy [29], and multiscale dispersion entropy [30,31], have been used to extract the characteristics of transformer fault signals. For example, Lu et.al. [29] used refine composite multiscale entropy (RCMSE) and time-frequency entropy (TFE) to extract the fault characteristics of transformer signals, and combined the improved kernel extreme learning machine (IKELM) to achieve the accurate identification of transformer faults. However, the above entropies rely too much on the selection of hyper-parameters, and the robustness of the entropy model parameters is poor, which requires significant time cost in parameter selection [32,33]. To overcome the above shortcomings of the entropies, Manis et al. [34] proposed a new signal complexity measurement tool—bubble entropy. It was proved by experiments that bubble entropy completely eliminated the influence of proportion factor and further reduced the importance of the embedding dimension, which is an entropy model with basically no parameters [35,36]. At the same time, in view of the fact that bubble entropy cannot comprehensively measure the complexity of time-series signals at multiple scales, inspired by the idea of time-shift, this paper proposes the time-shift multiscale bubble entropy, and uses TSMBE as the feature extraction tool to extract the fault characteristics of vibration signals in different states of power transformers.

Pattern recognition is an important part of power transformer fault diagnosis. It is the ultimate goal of transformer fault diagnosis to locate the rapid positioning of the transformer fault by identifying different state signals with classifiers. With the continuous development of artificial intelligence, machine learning-based models have been used for power load forecasting [37,38], power system security assessment [39,40] and circuit fault location [41,42]. The existing methods of transformer fault diagnosis often use machine learning algorithms such as support vector machine (SVM) [43,44], probabilistic neural network (PNN) [45] and back propagation neural network (BPNN) [46] as classifiers to effectively identify different transformer faults. Although the above methods can sometimes achieve good diagnostic results, problems such as difficulty of select hyper-parameters, easily falling into the local optimal solution, and a long training time, hinder their further application. As the latest model of stochastic parameter neural network [47], stochastic configuration network (SCN) overcomes the difficulty of hyper-parameter setting by virtue of its unique supervision mechanism and incremental network model. It has the advantages of low computational cost, high efficiency and not easily falling into the local optimal solution [48,49]. It has a wide range of application in chemical medicine [50], wind speed prediction [51] and the aviation industry [52]. In this paper, the features extracted by TSMBE are inputted into the SCN network to complete the pattern recognition of different state signals of the transformer.

Through the above analysis, the main innovations of this paper are as follows. Firstly, aiming to solve the problem that traditional multiscale entropy parameters are difficult to adjust, and inspired by the idea of time-shift, a nonlinear dynamic method with almost no hyper-parameters—TSMBE—is developed. TSMBE is then verified by experiments to have good feature extraction performance. Secondly, TSMBE is used as a feature extraction tool to perform the feature extraction of vibration signals in different states of transformers. The feature extraction effect of TSMBE under different hyper-parameters in actual transformer fault diagnosis is analyzed, and the results show that TSMBE is hardly affected by hyper-parameters. Then, SCN is introduced as the classifier of power transformer fault diagnosis, and a fault diagnosis method of power transformer based on TSMBE and stochastic configuration network is proposed to systematically identify different state signals of the transformer. Finally, by comparing the performance of different models in the measured transformer fault cases, it is concluded that the proposed method performs the best among all models, which verifies the superiority of the proposed method.

The rest of this paper is as follows: Section 2 introduces the relevant theoretical knowledge of the model and the performance verification process of the TSMBE algorithm; Section 3 refers the proposed model to actual transformer fault cases. Finally, Section 4 draws the conclusions of this study.

## 2. TSMBE-SCN Model

### 2.1. Bubble Entropy

Inspired by permutation entropy and Renyi entropy, Manis et al. [34] proposed a new tool for measuring time series complexity—bubble entropy. By eliminating the proportion factor and reducing the importance of embedding dimension, bubble entropy has stronger robustness to parameters and overcomes the shortcoming of traditional entropy parameter selection. The main steps of bubble entropy are as follows:
(1)Map time series $X={\left\{x_{i}\right\}}_{i=1}^{i=N}$ into m-dimensional space vector Z by phase space reconstruction:(1)$$Z=\left[\begin{array}{cccc} x\left(1\right) & x\left(1+d\right) & \cdot \cdot \cdot & x\left(1+\left(m-1\right)d\right)\\ \cdot \cdot \cdot & \cdot \cdot \cdot & \cdot \cdot \cdot & \cdot \cdot \cdot \\ x\left(j\right) & x\left(j+d\right) & \cdot \cdot \cdot & x\left(j+\left(m-1\right)d\right)\\ \cdot \cdot \cdot & \cdot \cdot \cdot & \cdot \cdot \cdot & \cdot \cdot \cdot \\ x\left(N-\left(m-1\right)d\right) & x\left(N-\left(m-2\right)d\right) & \cdot \cdot \cdot & x\left(N\right) \end{array}\right]$$
where m represents the embedding dimension, N is the length of time series, d is the delay time, and its value takes 1.

Use the bubble sort algorithm to sort each $Z_{m}\left(j\right)=\left\{x\left(j\right),x\left(j+d\right),\cdot \cdot \cdot ,x\left(j+\left(m-1\right)d\right)\right\}$ and calculate the exchange number n required for each vector, and then calculate the entropy $H^{m}$ of this distribution:(2)$$H^{m}=-\log\sum_{l=1}^{L}p_{l}^{2}$$
where L is the type number of different exchange numbers, $p_{l}$ represents the probability of different exchange numbers;
(2)Replace m with m+1 and repeat steps (1)–(2) to calculate $H^{m+1}$;(3)According to Formula (3), bubble entropy can be obtained:(3)$$BE\left(X,m,d\right)=\left(H^{m+1}-H^{m}\right)/\log\left(m+1/m-1\right)$$

### 2.2. Time-Shift Multiscale Bubble Entropy

Bubble entropy only measures the time complexity on a single scale and it is difficult to fully reflect the effective information of the signal. To solve this problem, Costa et al. [53] proposed the concept of multiscale entropy, which achieves the purpose of multiscale measurement of signal complexity by segmenting time series signals. In this paper, multiscale entropy and bubble entropy are combined, and TSMBE is proposed based on fractal theory, which overcomes the problems of insufficient coarse-grained degree of traditional multiscale entropy and difficult adjustment of parameters. The calculation process of TSMBE is as follows:
(1)As shown in Figure 1, segment the time series signal of length N into k subsequences through Formula (4):(4)$$Y_{k}^{\beta }=\left\{x_{\beta },x_{\beta +k},x_{\beta +2k},\cdot \cdot \cdot ,x_{\beta +k\left\lfloor \left(N-\beta \right)/k\right\rfloor }\right\}$$
where x is the sample point of the original signal, k represents the number of segmented subsequences, and $Y_{k}^{\beta }$ is the βth subsequence.(2)Calculate the bubble entropy of time series signals at all scales, and define the mean value of these attention entropy as the TSMBE at this scale k:(5)$$\mathrm{TSMBE}\left(k\right)=\frac{1}{k}\sum_{\beta =1}^{k}BE(Y_{k}^{\beta },m,d)$$(3)Use Formula (5) to calculate the TSMBE value of all scale factors τ, and take the set of these values as TSMBE. In this paper, τ is set to 10. The specific calculation process is shown in Figure 2:

#### 2.2.1. Parameter Discussion of TSMBE

The selection of parameters directly leads to the change of the TSMBE entropy value. In order to explore the influence of hyper-parameters on the entropy distribution of TSMBE, Gaussian white noise (GWN) and 1/*f* noise (FN) are introduced for simulation experiments (see Figure 3).

This paper analyzes the TSMBE and multiscale bubble entropy (MBE) value distributions of GWN and FN under different embedding dimensions, and the results are shown in Figure 4. As shown in Figure 4, the volatility of TSMBE entropy is weaker than that of MBE, indicating that changing the coarse-grained processing method can effectively improve the stability of MBE. At the same time, by comparing the distribution of TSMBE and MBE under different m, it is concluded that when m is greater than 5, the entropy values of TSMBE and MBE fluctuate to some extent. When m is less than 4, the calculated entropy values of TSMBE and MBE deviate from the entropy values under other m conditions. In addition, if m is too large, it will have high computational time costs. For comprehensive consideration, the hyper-parameter m of TSMBE algorithm is set to 4 in this paper.

#### 2.2.2. Performance Analysis of TSMBE

In this paper, the performance of TSMBE is comprehensively considered from the robustness of algorithm timing length and the recognition ability of different types of signals.

In order to test the robustness of the timing length of TSMBE, this paper analyzes the entropy value distribution of TSMBE and MBE under 50 groups of noises with different lengths of time series (N=256, N=512, N=1024, N=2048, N=4096, and N=8192). Additionally, the paper evaluates the robustness of the timing length of the algorithm by the change of mean value, with the specific results shown in Figure 5. It can be seen from the figure that, compared with MBE, the entropy distribution of TSMBE is smoother at different timing lengths. Among them, the maximum fluctuations of the mean MBE entropy value under different length noises (GWN and FN) reach 0.201 and 0.25, while the maximum fluctuations of the TSMBE entropy value are 0.138 and 0.23, indicating that the TSMBE algorithm is less affected by the length of time series. At the same time, comparing the entropy value distribution of time series signals with different lengths, the entropy value of time series signals with N less than 1024 deviates to some extent. It indicates that the stability of TSMBE and MBE is weakened when N is less than 1024, which requires that the length of time series signals must be greater than or equal to 1024.

In essence, TSMBE is a feature extraction tool, and whether it can effectively distinguish different types of signals becomes an important indicator of measuring its performance. The paper analyzes the recognition effects of TSMBE and MBE for different noises (GWN and FN signals with lengths of 2048), and measures the classification performance of different multiscale entropies through the entropy distribution and coefficient of variation (CV) value, with the results shown in Figure 6. As shown in Figure 6, the CV value of the multiscale entropy processed by the time-shift segmentation method is generally lower than that of the traditional multiscale entropy, indicating that the multiscale entropy obtained on the basis of the time-shift segmentation method is more stable. It indirectly verifies that the use of the time-shift segmentation method can effectively overcome the shortcomings of traditional multiscale entropy coarse-graining. However, by observing the distribution of TSMSE entropy values, it can be seen that the entropy value of TSMSE is not defined in high-scale cases, indicating that the sample entropy is greatly affected by timing length, and cannot adapt to a short time series. At the same time, comparing the entropy distribution of TSMBE and MBE, it can be seen that TSMBE has no crossover phenomenon at any scale, indicating that it has good signal recognition ability. In addition, comparing the CV values of the four multiscale entropies, the maximum CV value of TSMBE is only 0.497, which is far lower than that of the other three multiscale entropies, which verifies its good stability. According to the above analysis, it can be concluded that TSMBE has good recognition ability for different types of signals.

### 2.3. Stochastic Configuration Network

As a new type of random weight neural network with supervision mechanism, SCN is gradually constructed according to the supervision mechanism, which is different from the conventional feedforward neural network. The mechanism constrains the specific value range of random input weights and deviations. This supervision mechanism guarantees the general approximation properties of the SCN model generated by a given nonlinear mapping. The specific detailed process of SCN is described as follows:
(1)Use sigmoid as the activation function to calculate the output $g_{s}$ of the sth hidden node, when the hidden layer node is S−1, then the output $Z_{S-1}$ of the SCN is:(6)$$Z_{S-1}=\sum_{s=1}^{S-1}\gamma_{s}g_{s}(\omega_{s}^{T}W+b_{s})(S=2,3,4,\ldots L_{\max},Z_{0}=0)$$
where $\gamma_{s}$ represents the output weight of the sth hidden node, W is the input vector, and $L_{\max}$ is the maximum number of hidden layer nodes. At the same time, the network error $e_{S-1}$ can be calculated according to Formula (7):(7)$$e_{S-1}=Z-Z_{S-1}=[e_{S-1,1},e_{S-1,2},\cdots ,e_{S-1,D}]$$
(2)Introduce a supervision mechanism, and randomly assign the input weight and bias of node S through Formula (8):(8)$$\zeta ={\langle e_{S-1,h},g_{S}\rangle }^{2}-b_{g}^{2}(1-r-\mu_{S}){\left\|e_{S-1,h}\right\|}^{2}\geq 0,h=1,2,\cdots ,D$$
where $0\leq \left\|g\right\|<b_{g}$,$b_{g}\in R^{+}$; r represents the regularization parameter, ranging from 0 to 1; and $\mu_{S}=(1-r)/S+1$. The weight and bias corresponding to the maximum value ζ obtained by repeating *T*_max_ experiment are the required values.(3)Use the least square method combined with weight ω and bias b to calculate the hidden layer output weight:(9)$$[\gamma_{1},\gamma_{2},\cdots ,\gamma_{L}]=\arg\min{\left\|Z-\sum_{j=1}^{L}\gamma_{j}g_{j}\right\|}^{2}$$

Continue to increase the hidden layer nodes, and combine the given γ, ω, and b with Formula (8) to achieve the smallest error value, and finally output the optimal model. At the same time, according to the SCN hyper-parameter setting principle in [48], the proportion factor λ of the input weight and bias is set to {0.5, 1, 5, 10, 30, 50, 100, 150, 200, 250}, this value of $T_{\max}$ is 100; The regularization parameter r is set to {0.9, 0.99, 0.9999, 0.99999, 0.999999}, and the allowable error ε is set to 0.001. The specific calculation process of SCN is shown in Figure 7.

### 2.4. Transformer Fault Diagnosis Model Based on TSMBE-SCN

The overall process of transformer fault diagnosis is shown in Figure 8, and the TSMBE value of vibration signal is extracted as the feature vector. At the same time, the feature sample data are divided into training set and test set. Then, the SCN model is trained by the training set samples, and the recognition performance of the diagnostic model is tested by the test set samples.

## 3. Experimental Case Analysis

### 3.1. Experimental Platform and Data Description

In order to verify the effectiveness and superiority of the proposed diagnosis model, this paper studied the vibration signals of the inter-turn short circuit, under-excitation and over-excitation faults of the transformer in [54] as the research objects.

In order to avoid information leakage caused by repeated sampling, as shown in Figure 9, the experiment used the data sample length of 1024 as the standard, and adopted the non-overlapping sampling method to segment the measured vibration signal. By adjusting the fault current and voltage, a total of 510 groups of six kinds of fault signals were collected, including under-excitation, over-excitation and inter-turn short circuit. Each state sample had 85 groups, and the sampling frequency was 30 kHz. In addition, each group of signals was normalized. The vibration signal waveform is shown in Figure 10.

Figure 10 shows the waveforms of six state signals, in which the excitation fault vibration signals were generated by regulating the voltage. In this paper, signals generated at 320 V, 360 V, 400 V and 440 V were selected to simulate the under-excitation signal I, under-excitation signal II, normal excitation signal and over-excitation signal. Vibration signals under 5 A and 10 A, two different current conditions, were selected to simulate the inter-turn short circuit signal I and inter-turn short circuit signal II, respectively. In addition, it was impossible to effectively distinguish the fault signals of transformers in different states by naked eye, and the signal feature information contained in the waveform signal was required to be deeply excavated. To facilitate memory, the six different signals were abbreviated as ‘NE’, ‘UE1’, ‘UE2’, ‘OE’, ‘TSC1’ and ‘TSC2’.

### 3.2. Fault Feature Extraction of Transformer Vibration Signal

TSMBE was used as a feature extraction tool for transformer vibration signal feature extraction. MBE [55], multiscale dispersion entropy (MDE) [30], time-shift multiscale dispersion entropy (TSMDE) [56], multiscale sample entropy (MSE) [53] and time-shift multiscale entropy (TSMSE) [57] were introduced for comparative experiments. All models were implemented on Matlab 2019b platform, and the parameter settings of different multiscale entropy are shown in Table 1.

Multiscale entropy such as TSMBE was used to extract fault features of vibration signals in different states, and t-distributed stochastic neighbor embedding (T-SNE) was used to perform extracted features visualization. The performance of feature extraction tool was measured by observing the distribution of vibration signal features in different states, and specific results are shown in Figure 11. It can be seen from Figure 11 that only sporadic ‘TSC1’ and ‘TSC2’ of the fault features extracted by TSMBE are mixed, indicating that TSMBE has good feature extraction performance, while the fault features extracted by the other five multiscale entropies had certain mixed phenomena. For example, the ‘UE1’, ‘UE2’ and ‘NE’, in the features extracted by MSE, had large aliasing, and MSE could not identify the ‘TSC1’ and ‘TSC2’ signals, while TSMSE had low feature extraction performance due to the limitation of timing length, and its feature extraction performance was not as good as MSE in short time series signals. In addition, compared with MDE and TSMDE, it was concluded that the time-shift segmentation method, instead of the traditional coarse-grained method, could indeed improve the feature performance of the model to a certain extent. The main feature of TSMDE extraction is that ‘TSC1’ and ‘TSC2’ have large aliasing. The features extracted by MBE also have large aliasing, which further shows that the simple coarse-grained segmentation method cannot effectively adapt to the feature extraction of complex fault signals of transformers.

The above analysis shows that, compared with the other five multiscale entropy models, the transformer feature vector extracted by TSMBE had the best distinguishing effect, and only sporadic signals were aliased, which verifies that TSMBE has good feature extraction performance.

### 3.3. Pattern Recognition of Transformer Signal

The features extracted by multiscale entropy were inputted into the SCN model to complete the pattern recognition of vibration signals of different transformer states. However, the maximum number of nodes in the hidden layer is an important parameter that affects network performance. This paper determined the optimal parameter value by analyzing the recognition effect of SCN under different $L_{\max}$. As shown in Figure 12, the paper used the features extracted by TSMBE as feature vectors, using the 2-fold cross-validation method to divide the training set and the test set, and analyzed the SCN diagnosis with $L_{\max}$ within 10 within 100. Each group of experiments was repeated 20 times, and the average diagnostic rate and standard deviation were used to measure the diagnostic performance of the model. The analysis showed that when $L_{\max}$ was 30, SCN achieved a diagnostic rate of 99.01% and a standard deviation of 0.634%, and the diagnostic effect was higher than other $L_{\max}$ models. Therefore, the $L_{\max}$ of SCN was set to 30.

In order to analyze the specific diagnosis of different models, the sample set was divided approximately equally into a training set and test set, and the diagnosis results of the different diagnostic models were analyzed through the confusion matrix diagram. Specific results are shown in Figure 13. As shown in Figure 13, the proposed method only misjudged the two types of signals ‘TSC1’ and ‘TSC2’, and two ‘TSC2’ signals were misjudged as ‘TSC1’, which was consistent with the results of feature extraction. Limited by the length of time, the diagnostic effect of TSMSE was the most unsatisfactory and was almost impossible to identify any state signal. By comparing the confusion matrix diagrams of MDE-SCN and TSMDE-SCN, MBE-SCN and TSMBE-SCN comparison models, it was concluded that compared with the MDE-SCN and MBE-SCN models obtained by traditional coarse-grained partitioning method, TSMDE-SCN and TSMBE-SCN showed excellent diagnostic performance. It indicated that the time-shift segmentation method could improve the diagnostic effect of the model to some extent. By comparing the MDE-SCN and MBE-SCN, TSMDE-SCN and TSMBE-SCN models, it was concluded that the bubble entropy-based multiscale entropy had better diagnostic performance.

In order to avoid the influence of random experiments on the final experimental results, and to verify the general performance of the proposed method, this paper divided the training set and the test set by the cross-validation method. Each group of experiments was repeated independently 20 times, and the mean value and standard deviation were used to measure the performance of different diagnostic models; the specific results are shown in Figure 14. As shown in Figure 14, compared with the other five diagnostic models, the proposed method had remarkable advantages in both diagnostic accuracy and algorithm stability. Under different folding numbers (K=2,3,4,5,6 and 7), its diagnostic mean values were 99.01%, 99.1%, 99.11%, 99.11%, 99.14% and 99.02%, and the standard deviations of diagnostic rates were 0.665%, 0.762%, 0.971%, 0.882%, 0.985% and 1.279%. TSMSE-SCN was affected by the timing length, and the diagnostic model performed poorly. The mean diagnostic rate was lower than 70% in all experiments, indicating that the model was not competent for the transformer in this paper. At the same time, comparing the diagnostic rates of other time-shift multiscale entropy and traditional multiscale entropy diagnostic models, the results showed that the multiscale entropy diagnostic model based on time-shift segmentation had better diagnostic effect. It showed that the new time series segmentation method could deeply mine fault information, overcome the shortcomings of traditional multiscale entropy coarse-grained, and further improve the diagnostic performance of the model.

The recognition performance of the classifier is one of the important factors that determine the transformer fault diagnosis. BPNN, extreme learning machine (ELM), and SVM, were introduced for comparative experiments. The hyper-parameters of BPNN were set as follows: the topological structure was 10-10-6, the number of iterations was 1000, the learning rate was 0.05, and the training target was 0.00001. The hyper-parameters of SVM were set as follows: the penalty factor was set to 1, the kernel function selected the ‘RBF’ function, and the kernel function parameter was set to 1/*C* (*C* is the number of categories, which was set to 1/6 in this paper). The hyper-parameters of ELM were set as follows: the number of hidden nodes was set to 100. The features extracted by TSMBE were used as feature vectors, which were divided into a training set and test set according to the cross-validation method, and inputted into four machine learning algorithms for recognition. Each group of experiments was repeated independently 20 times. The specific results are shown in Figure 15. It can be clearly seen from the figure that the diagnostic effect of BPNN was far inferior to the other three classifiers, and the proposed model showed the best diagnostic effect in all folding numbers. For example, compared with the other three diagnostic models (TSMBE-ELM, TSMBE-SVM and TSMBE-BPNN), the diagnostic rate of TSMNE-SCN increased by 1.17%, 2.66% and 15.01%, respectively, under folding number 2, while the standard deviation decreased by 0.05%, 0.894% and 7.906%, respectively. Through the above analysis, it was concluded that SCN is a stable and effective classifier.

In addition, in order to explore whether the hyper-parameter *m* will have a huge impact on the TSMBE, the paper analyzed the diagnostic effect of TSMBE-SCN under different embedding vectors (m=2,3,4,5 and 6); specific results are shown in Figure 16. It can be seen from Figure 16 that the diagnostic mean value and standard deviation of TSMBE-SCN under different parameters m were roughly equal. The fluctuation of the mean value was not more than 0.28%, and the fluctuation of the standard deviation was not more than 0.62%. All models achieved a diagnostic mean value of more than 98.73% and a standard deviation of less than 0.728%, indicating that the hyper-parameter *m* had little effect on the final experimental results. It also proved that TSMBE is an algorithm that is not affected by the hyper-parameter from the experimental perspective.

## 4. Conclusions

In this paper, a fault diagnosis method combining time-shift multiscale bubble entropy and stochastic configuration network was proposed to achieve the early fault diagnosis of transformer faults in different states. The following conclusions were obtained by experimental verification:(1)Aiming to solve the shortcomings of insufficient coarse-grained scale, and the difficulty in determining hyper-parameters of traditional multiscale entropy, a new nonlinear dynamic method—time-shift multiscale bubble entropy—was developed. The hyper-parameters and the applicable shortest timing length of the TSMBE algorithm were determined through simulation experiments, and comparison experiments of TSMBE, MBE, TSMSE and MSE algorithms were carried out to verify that TSMBE had good signal recognition performance and robustness of timing length;(2)The proposed method was applied to real transformer fault cases. Compared with MBE-SCN, TSMSE-SCN, MSE-SCN, TSMDE-SCN and MDE-SCN models, the proposed model performed best. The diagnostic mean values under different folding numbers were 99.01%, 99.1%, 99.11%, 99.11%, 99.14% and 99.02%, respectively, and the diagnostic standard deviations were 0.665%, 0.762%, 0.971%, 0.882%, 0.985% and 1.279%, respectively, which proved the superiority of the proposed method;(3)Comparing the recognition performance of the four classifiers SCN, BPNN, SVM and ELM, it was concluded that the diagnostic rate of TSMNE-SCN increased by 1.17%, 2.66% and 15.01%, while the standard deviation decreased by 0.05%, 0.894% and 7.906%, indicating that SCN had the best recognition performance. In addition, the diagnostic results of the proposed method under different embedding dimensions *m* were discussed. It showed that the mean value of different models did not fluctuate more than 0.28%, and the standard deviation did not fluctuate more than 0.62%. All models achieved a diagnostic mean value of more than 98.73%, and the standard deviation was also less than 0.728%, indicating that TSMBE was an algorithm not affected by parameters.

This paper developed a new transformer early fault diagnosis method, which had a good guiding role. However, the proposed method only relied on a single direction vibration signal to perform the fault diagnosis of the transformer. In actual production experiments, problems such as difficulty in selecting measurement points and incomplete reflection of fault information may be encountered. Therefore, in the next work, the authors hope to extend the TSMBE algorithm into the field of multi-channel fault feature extraction, and propose the use of time-shift multivariate multiscale bubble entropy to achieve the joint diagnosis of multi-channel signals of a transformer.

## Figures and Tables

**Figure 1 entropy-24-01135-f001:**
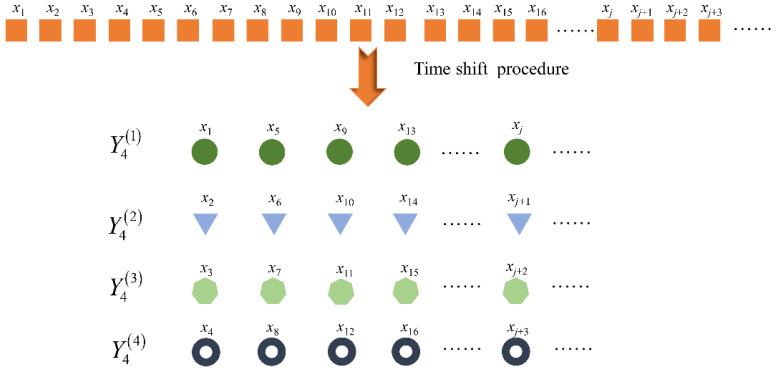
Time shift segmentation process under k=4.

**Figure 2 entropy-24-01135-f002:**
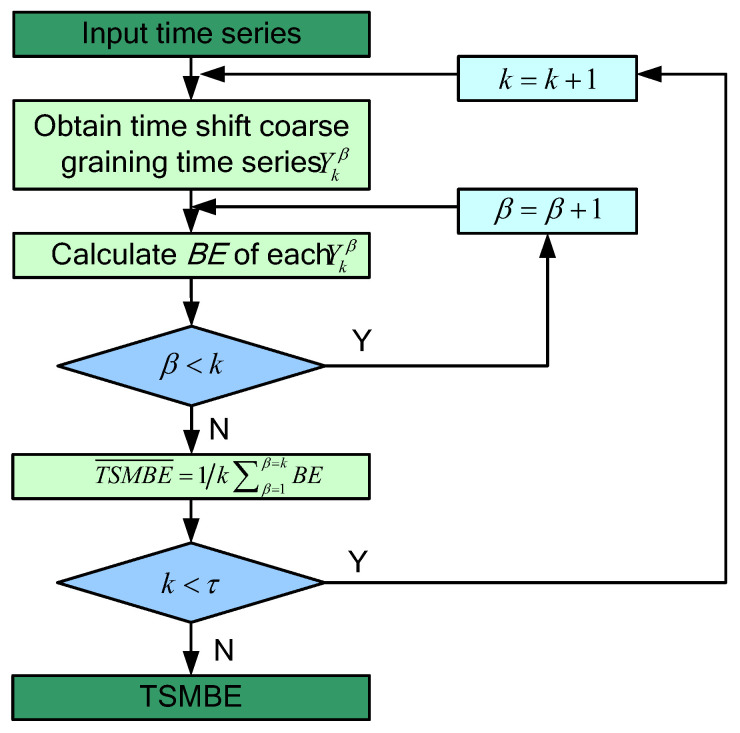
The calculation process of TSMBE.

**Figure 3 entropy-24-01135-f003:**
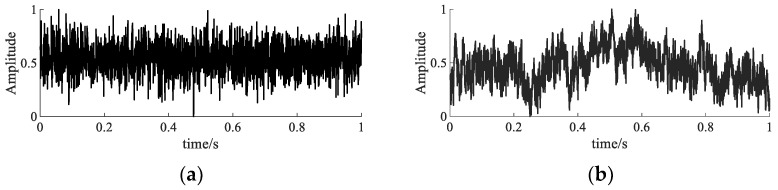
The vibration signal of noise. (**a**) The vibration signal of GWN. (**b**) The vibration signal of FN.

**Figure 4 entropy-24-01135-f004:**
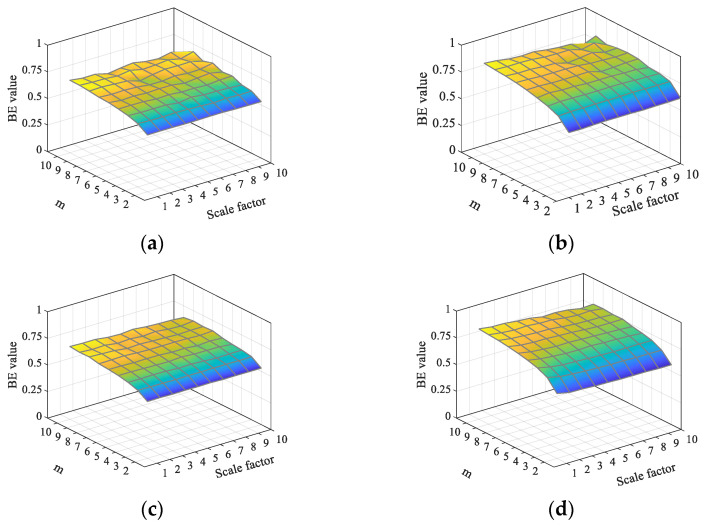
Distribution of entropy under different embedding dimensions. (**a**) MBE distribution of GWN. (**b**) MBE distribution of FN. (**c**) TSMBE distribution of GWN. (**d**) TSMBE distribution of FN.

**Figure 5 entropy-24-01135-f005:**
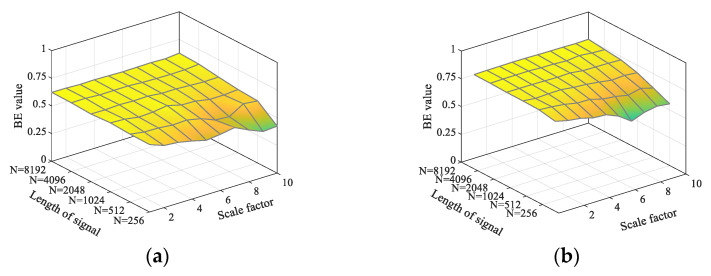
Entropy distribution under different lengths of noise signal. (**a**) MBE distribution under different lengths (GWN). (**b**) MBE distribution under different lengths (FN). (**c**) TSMBE distribution under different lengths (GWN). (**d**) TSMBE distribution under different lengths (FN).

**Figure 6 entropy-24-01135-f006:**
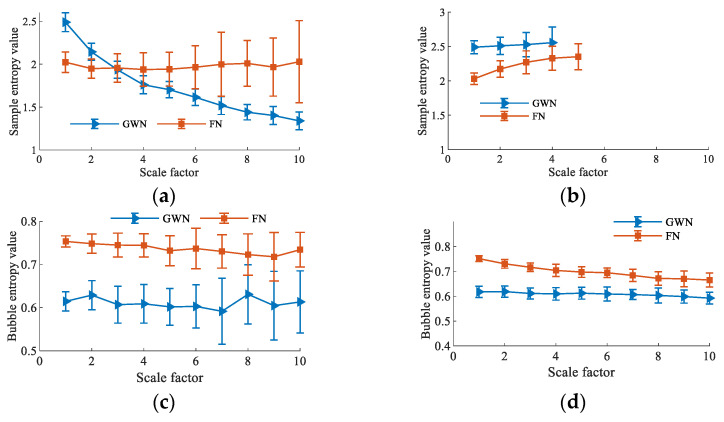
Recognition of multiscale entropy under different types of noise. (**a**) MSE distribution under different types of noise. (**b**) TSMSE distribution under different types of noise. (**c**) MBE distribution under different types of noise. (**d**) TSMBE distribution under different types of noise. (**e**) CV value by MSE under different types of noise. (**f**) CV value by TSMSE under different types of noise. (**g**) CV value by MBE under different types of noise. (**h**) CV value by TSMBE under different types of noise.

**Figure 7 entropy-24-01135-f007:**
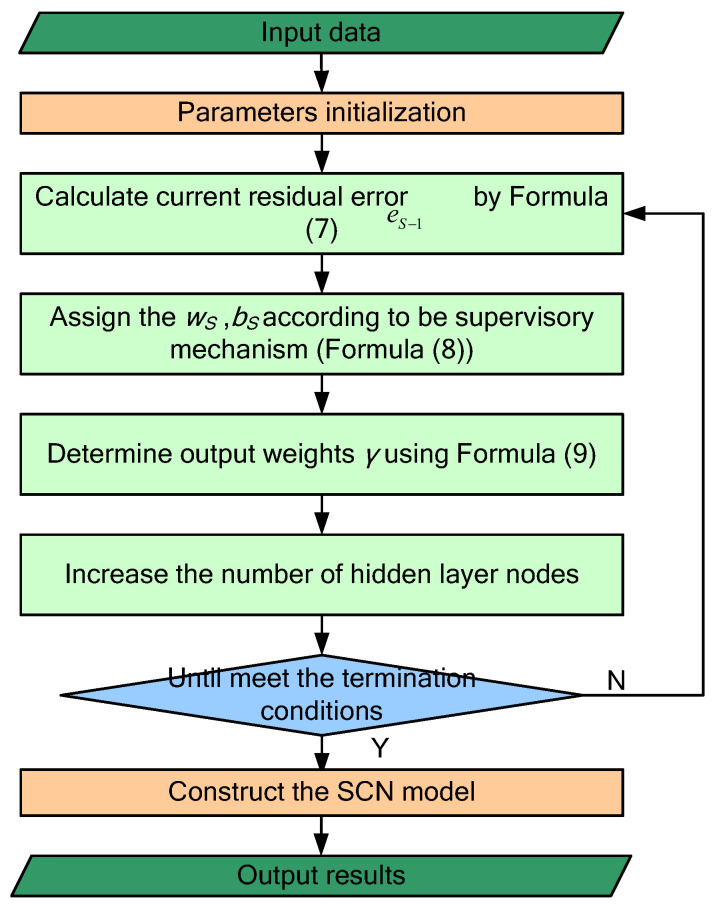
The flowchart of stochastic configuration network.

**Figure 8 entropy-24-01135-f008:**
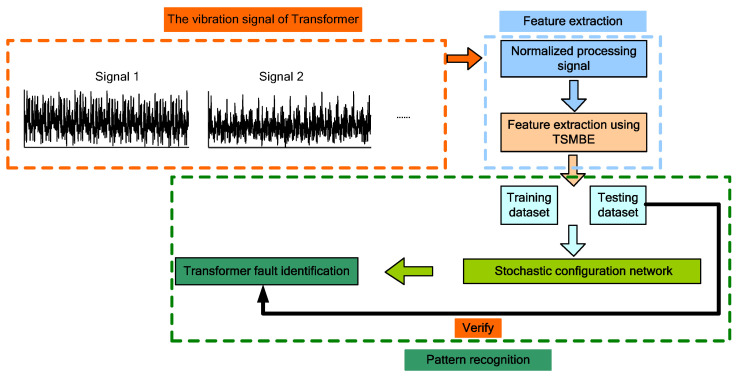
Flow chart of the proposed method.

**Figure 9 entropy-24-01135-f009:**
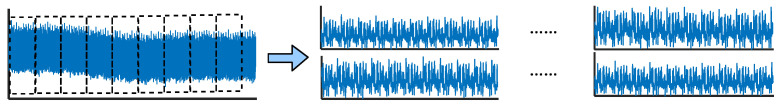
Non-overlapping sampling method.

**Figure 10 entropy-24-01135-f010:**
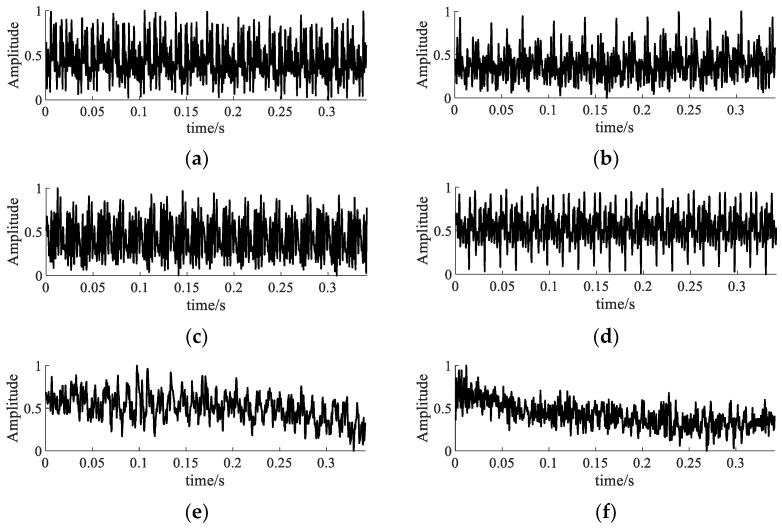
Waveform diagram of measured signal of transformer. (**a**) The signal of normal excitation. (**b**) The signal of under-excitation (type I). (**c**) The signal of under-excitation (type II). (**d**) The signal of over-excitation. (**e**) The signal of turn-to-turn short circuit (type I). (**f**) The signal of turn-to-turn short circuit (type II).

**Figure 11 entropy-24-01135-f011:**
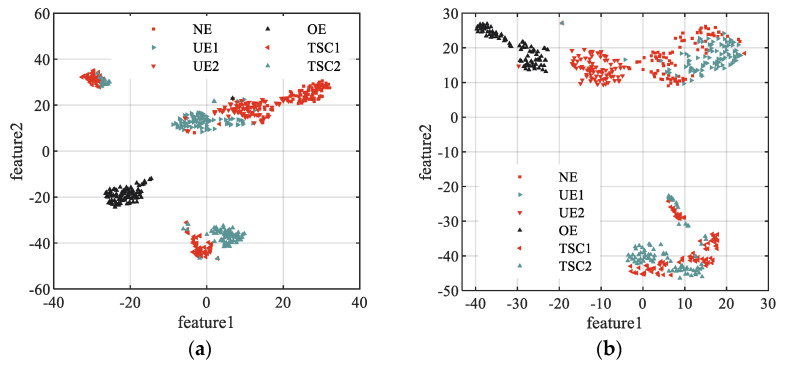
The visualization results of model feature. (**a**) The visualization results of MSE. (**b**) The visualization results of MDE. (**c**) The visualization results of MBE. (**d**) The visualization results of TSMSE. (**e**) The visualization results of TSMDE. (**f**) The visualization results of TSMBE.

**Figure 12 entropy-24-01135-f012:**
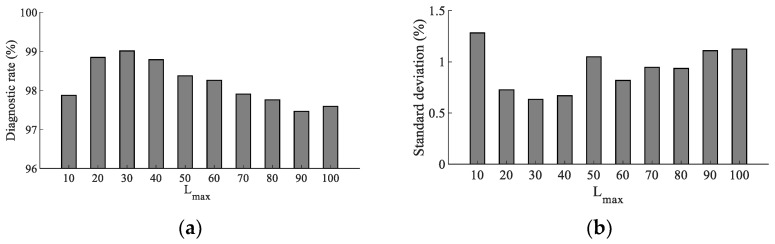
Diagnostic mean value of SCN model under different $L_{\max}$. (**a**) Diagnostic mean value of SCN. (**b**) Diagnostic mean value of SCN model under different $L_{\max}$ model under different $L_{\max}$.

**Figure 13 entropy-24-01135-f013:**
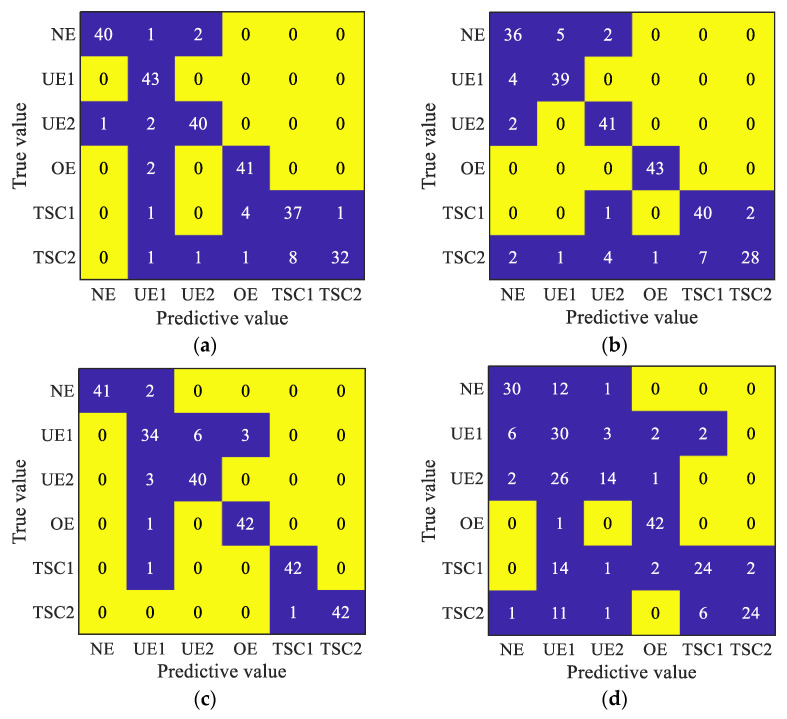
Model confusion matrix diagram. (**a**) The confusion matrix of MSE-SCN. (**b**) The confusion matrix of MDE-SCN. (**c**) The confusion matrix of MBE-SCN. (**d**) The confusion matrix of TSMSE-SCN. (**e**) The confusion matrix of TSMDE-SCN. (**f**) The confusion matrix of TSMBE-SCN.

**Figure 14 entropy-24-01135-f014:**
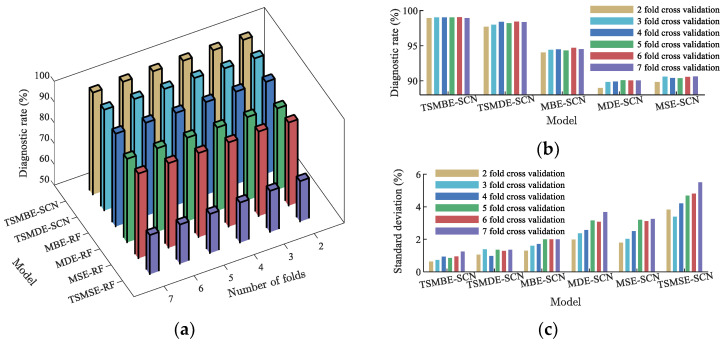
Fault diagnosis of models under different fold numbers. (**a**) Diagnostic mean of models. (**b**) Diagnostic mean of main models. (**c**) Diagnostic standard deviation of models.

**Figure 15 entropy-24-01135-f015:**
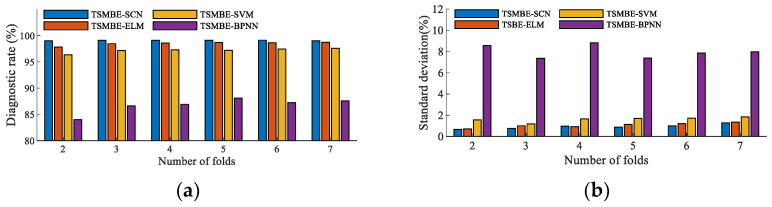
Fault diagnosis of classifiers under different fold numbers. (**a**) Diagnostic mean value of different classifiers. (**b**) Diagnostic standard deviation of different classifiers under different fold numbers under different fold numbers.

**Figure 16 entropy-24-01135-f016:**
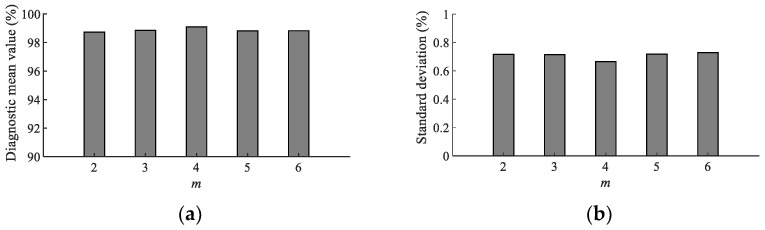
Fault diagnosis of model under different fold numbers. (**a**) Diagnostic mean value under different *m*. (**b**) Diagnostic standard deviation under different *m*.

**Table 1 entropy-24-01135-t001:** The parameter settings of different multiscale entropy model.

Model	Parameter				
	*m*	*d*	Number of Categories *c*	Threshold *r*	Scale Factor *τ*
TSMBE	4	1			10
TSMDE	2	1	5		10
TSMSE	2	1		0.15×SD	10
MBE	4	1			10
MDE	4	1			10
MSE	2	1	5	0.15×SD	10

## Data Availability

Not applicable.

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
