# Peer review of "Fault Diagnosis of Power Transformer Based on Time-Shift Multiscale Bubble Entropy and Stochastic Configuration Network"

_entropy, 2022, doi:10.3390/e24081135_

Round 1
Reviewer 1 Report
The paper is well written and well organized. Some minor corrections are needed:
- the following acronyms are not explained: CV, CMBE, BE and SE in fig.6, ELM
- the bibliography [34] is not called in the body of the paper
. the sections numbering method must be made more uniform
- two different acronyms are used for the same quantity (GWN and WGN). Please select one of the two and use always that
- in line 294 the reference to the equation should likely be 7 and not 8
- in line 262 the step 3 is missing
-in line 470 ELM is mentioned twice whereas BPNN is missing
Reviewer 2 Report
This work is interesting and provides nice results. The reviewer thinks that there are comments on the research itself. Instead, adding several recent works covering the deep-learning-based fault diagnosis of power transformers in reference will help the readers to follow the research stream. The following papers are recommended to be cited:
[R1] Kim, Sunuwe, et al. "A semi-supervised autoencoder with an auxiliary task (SAAT) for power transformer fault diagnosis using dissolved gas analysis." IEEE Access 8 (2020): 178295-178310.
[R2] Kim, Sunuwe, et al. "Learning from even a weak teacher: Bridging rule-based Duval method and a deep neural network for power transformer fault diagnosis." International Journal of Electrical Power & Energy Systems 136 (2022): 107619.
[R3] Seo, Boseong, et al. "Missing data imputation using an iterative denoising autoencoder (IDAE) for dissolved gas analysis." Electric Power Systems Research 212 (2022): 108642.
Reviewer 3 Report
Comments to authors
This work proposes a transformer fault diagnosis method based on the combination of time-shift multiscale bubble entropy (TSMBE) and stochastic configuration network (SCN). Firstly, bubble entropy is introduced to overcome the shortcomings of traditional entropy models that rely too much on hyperparameters. Secondly, a tool for measuring signal complexity is proposed based on bubble entropy—TSMBE. Then, the TSMBE of the transformer vibration signal is extracted as a fault feature. Finally, the fault feature is input into the stochastic configuration network model to achieve accurate identification of different transformer state signals. But some comments should be taken into your consideration.
1- Editing English requires some effort to enhance your work.
2- Comparisons with recently published work are not considered in your results section. This manuscript's last part of your results section should include a comparison with recently published works.
